# RUN: Reversible Unfolding Network for Concealed Object Segmentation

**Chunming He** [1] , **Rihan Zhang** [1] , **Fengyang Xiao** [1] , **Chengyu Fang** [2] , **Longxiang Tang** [2] ,
**Yulun Zhang** [3] , **Linghe Kong** [3] , **Deng-Ping Fan** [4] , **Kai Li** [5] , **Sina Farsiu** [1]

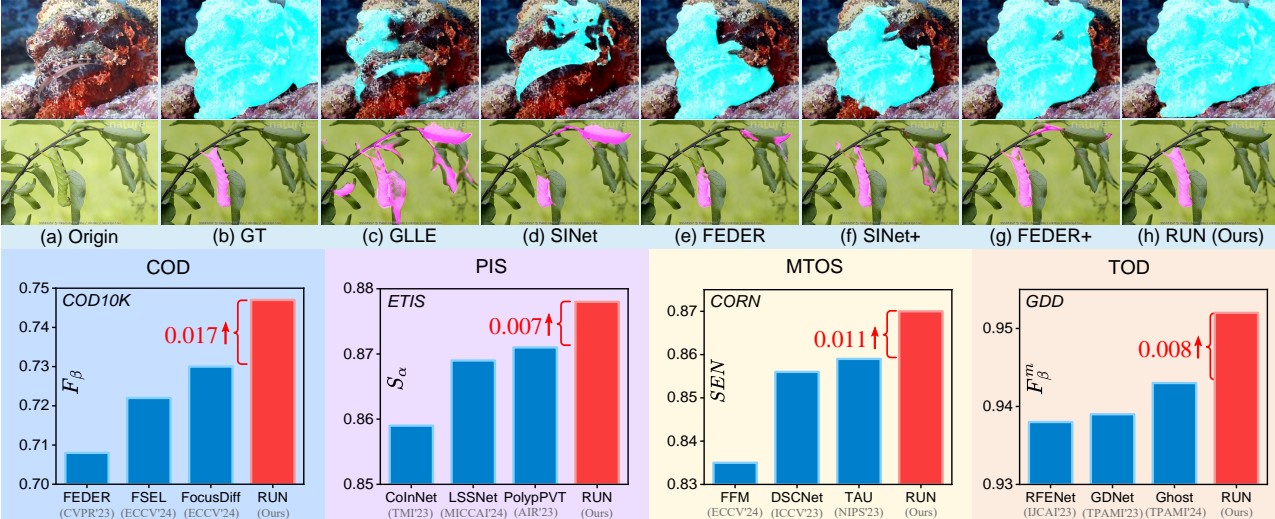

*Figure 1.* Results of existing COS methods, including GLLE (Wang et al., 2019), SINet (Fan et al., 2020a), and FEDER (He et al., 2023b). Our RUN demonstrates superiority in accurately segmenting concealed objects (in the top section) and achieves leading places across multiple COS tasks (in the bottom section): camouflaged object detection (COD), polyp image segmentation (PIS), medical tubular object segmentation (MTOS), and transparent object detection (TOD). In the top section, concealed object masks are highlighted in blue and pink, overlaid on the original images for visual clarity. FEDER+ and SINet+ indicate integrating FEDER and SINet with our RUN framework. For the bottom section, we employ commonly used datasets, methods, and metrics relevant to each task.

## Abstract

Existing concealed object segmentation (COS) methods frequently utilize reversible strategies, *e.g.*, background reversible attention and foreground-background reversible calibration, to address uncertain regions. However, these approaches are typically restricted to the mask domain, leaving the potential of the RGB domain underexplored. To address this, we propose the Reversible Unfolding Network (RUN), which applies reversible strategies across both mask and RGB domains through a theoretically grounded framework, enabling accurate segmentation. RUN first formulates a novel COS model

by incorporating an extra residual sparsity constraint to minimize segmentation uncertainties. The iterative optimization steps of the proposed model are then unfolded into a multistage network, with each step corresponding to a stage. Each stage of RUN consists of two reversible modules: the Segmentation-Oriented Foreground Separation (SOFS) module and the Reconstruction-Oriented Background Extraction (ROBE) module. SOFS applies the reversible strategy at the mask level and introduces Reversible State Space to capture non-local information. ROBE extends this to the RGB domain, employing a reconstruction network to address conflicting foreground and background regions identified as distortion-prone areas, which arise from their separate estimation by independent modules. As the stages progress, RUN gradually facilitates reversible modeling of foreground and background in both the mask and RGB domains, directing the network's attention to uncertain regions and mitigating false-positive and false-negative results. Extensive

[1]Duke University. [2]SIGS, Tsinghua University. [3]Shanghai Jiao Tong University. [4]Nankai Institute of Advanced Research (SHENZHEN-FUTIAN). [5]Meta. Correspondence to: Sina Farsiu <sina.farsiu@duke.edu>, Fengyang Xiao <fengyang.xiao@duke.edu>.

*Proceedings of the 42nd International Conference on Machine Learning*, Vancouver, Canada. PMLR 267, 2025. Copyright 2025 by the author(s).

experiments verify the superiority of RUN and highlight the potential of unfolding-based frameworks for COS. Code is available at https://github.com/ChunmingHe/RUN.

# 1. Introduction

Concealed object segmentation (COS) aims to segment objects that are visually blended with their surroundings. It serves as an umbrella term with various applications, including camouflaged object detection (He et al., 2024b), polyp image segmentation (He et al., 2025b), and transparent object detection (Xiao et al., 2023), among others.

COS is a challenging problem due to the intrinsic similarity between the object and its background. Traditional methods address this challenge by relying on manually designed models with hand-crafted feature extractors tailored to subtle differences in textures, intensities, and colors (Wang et al., 2019). While offering clear interpretability, they often structure in complex scenarios. Deep learning advances COS by leveraging its strong generalization capabilities, driven by powerful feature extraction mechanisms. Early learning-based approaches, such as SINet (Fan et al., 2020a), primarily focus on foreground regions for segmentation, often overlooking discriminative cues in the background, leading to suboptimal performance (see Fig. 1). Recent algorithms, such as FEDER (He et al., 2023b), have sought to refine segmentation masks by reversibly modeling both foreground and background regions at the mask-level.

Reversible modeling enhances the network's capacity to extract subtle discriminative cues by directing attention to uncertain regions—pixels with values that are neither 1 nor 0—thus improving segmentation results. However, current methods restrict the application of reversible strategies to the mask level, leaving the potential of the RGB domain underexplored. Such information can assist in identifying discriminative cues and enhancing segmentation quality. As shown in Fig. 2 (d) and (e), when reversibly separating the image into foreground and background regions based on the mask, the uncertainty regions in the mask tend to manifest as color distortion in the RGB space. Addressing these translates to a more precise separation of the foreground and background. In this case, two seemingly independent tasks—object segmentation and distortion restoration—share the same optimization goal. Existing research has shown that jointly optimizing such tasks helps guide the network toward an optimal solution (Xu et al., 2023).

To achieve this, we first introduce a deep unfolding network termed the Reversible Unfolding Network (RUN) for COS. RUN established a theoretical foundation to reversibly integrate the two aforementioned tasks, rather than directly combining them, to achieve more accurate segmentation. The COS task is formulated as a foreground-background separation process, and a new segmentation model is developed by incorporating a residual sparsity constraint to reduce segmentation uncertainties. The iterative optimization steps of the model-based solution are then unfolded into a multi-stage network, with each step corresponding to a stage. Each stage comprises two reversible modules: the Segmentation-Oriented Foreground Separation (SOFS) module and the Reconstruction-Oriented Background Extraction (ROBE) module. By integrating optimization solutions with deep networks, our RUN framework achieves an effective balance between interpretability and generalizability.

We implement the reversible strategy within the mask domain in SOFS and within the RGB domain in ROBE. In SOFS, the mask is initially updated strictly according to the optimization solution. Subsequently, the Reversible State Space (RSS) module, recognized for its strong capacity to extract non-local information, is employed to refine the segmentation mask using the previously estimated mask and background. In ROBE, the process begins with a mathematical update of the background. A lightweight network is then used to reconstruct the entire image, while simultaneously refining the background, based on the estimated foreground and background. Since the estimation of foreground and background regions is performed by distinct modules, their assessments of concealed content can differ (see Fig. 2 (d) and (f)). Regions of conflicting judgments are identified as distortion-prone areas during the reconstruction process (see Fig. 2 (g) and (h)). This auxiliary reconstruction task, which targets the resolution of such distortions, effectively directs the network's attention to challenging regions where distinguishing between foreground and background is particularly difficult, improving segmentation performance.

As the stages progress, RUN incrementally facilitates reversible modeling of foreground and background in both the mask and RGB domains. This approach effectively focuses the network on uncertain regions, reducing false-positive and false-negative outcomes. Notably, RUN exhibits high flexibility, allowing seamless integration with existing methods to achieve further performance enhancements.

Our contributions are summarized as follows:

(1) We propose RUN for the COS task. To the best of our knowledge, this represents the first application of a deep unfolding network to address the COS problem, thereby balancing interpretability and generalizability.

(2) RUN proposes a novel COS model designed to reduce segmentation uncertainties and introduces SOFS and ROBE modules to integrate model-based optimization solutions with deep networks. By enabling reversible modeling of foreground and background across both the mask and RGB

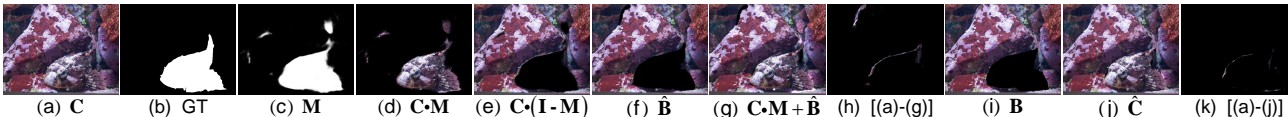

| (a) **C** | (b) GT | (c) **M** | (d) **C·M** | (e) **C·(I - M)** | (f) **B̂** | (g) **C·M+B̂** | (h) [(a)-(g)] | (i) **B** | (j) **Ĉ** | (k) [(a)-(j)] |

*Figure 2.* Correspondence between uncertainties in the mask domain and distortions in the RGB domain. **C** is the concealed image and **B̂** is the estimated background, which has conflicting judgments of concealed regions with the mask **M**. This conflict leads to distortion-prone areas in their direct combination (g). Panel (h) illustrates the difference between (g) and the original image (a). However, after refinement through the network $\mathcal{B}(\cdot)$, the reconstructed image **Ĉ** becomes much closer to the original image, accompanied by a refined background **B** with improved accuracy. This refined background is passed to the next stage to further facilitate segmentation.

domains, RUN directs the network's focus to uncertain regions, reducing false-positive and false-negative outcomes.

(3) Experiments on five COS tasks, as well as salient object detection, validate the superiority of our RUN method. Besides, its plug-and-play structure underscores the effectiveness and adaptability of unfolding-based frameworks for the COS task and other high-level vision tasks.

## 2. Related Works

**Concealed object segmentation**. Deep learning methods have advanced COS (Xiao et al., 2024). Among them, those using reversible techniques to segment from foreground and background aspects are gaining attention. PraNet (Fan et al., 2020b) introduced a parallel structure with reversible attention to enhance segmentation. FEDER (He et al., 2023b) used foreground and background masks to identify concealed objects with edge assistance. BiRefNet (Zheng et al., 2024) proposed a reconstruction module to refine the mask with gradient information. However, they only focus on the mask level, leaving the RGB domain underexplored. Hence, we propose the first deep unfolding network, RUN, for COS. RUN proposes a novel COS model and introduces SOFS and ROBE. By integrating optimization solutions with deep networks, RUN enables reversible modeling across mask and RGB domains, improving segmentation accuracy.

**Deep unfolding network**. The deep unfolding network, a well-established technique in low-level vision, integrates model-based and learning-based approaches (He et al., 2023a; Fang et al., 2024), offering enhanced interpretability compared to pure learning-based methods. However, its application in high-level vision remains underexplored, primarily due to the lack of explicit intrinsic models for high-level vision tasks. In this paper, we introduce a deep unfolding network, RUN, in COS and formulate a novel COS model. RUN achieves more accurate segmentation results by integrating optimization-based solutions with deep networks, verifying its potential for advancing COS.

## 3. Methodology

### 3.1. COS Model

A concealed image **C** can be decomposed into its foreground region **F** and background region **B**, expressed as

$$\mathbf{C} = \mathbf{F} + \mathbf{B}. \tag{1}$$

Based on Eq. (1), the foreground and background regions can be obtained by optimizing the objective function:

$$L(\mathbf{F}, \mathbf{B}) = \frac{1}{2}\|\mathbf{C} - \mathbf{F} - \mathbf{B}\|_2^2 + \beta\varphi(\mathbf{F}) + \lambda\phi(\mathbf{B}), \tag{2}$$

where $\|\cdot\|_2$ is a $\ell_2$-norm for smooth. $\varphi(\mathbf{F})$ and $\phi(\mathbf{B})$ are regularization terms for **F** and **B** with two trade-off parameters $\beta$ and $\lambda$. To suit the segmentation task, we directly focus on the mask **M**, where $\mathbf{F} = \mathbf{C} \cdot \mathbf{M}$, and · is dot product. Substituting into Eq. (2), the objective function becomes

$$L(\mathbf{M}, \mathbf{B}) = \frac{1}{2}\|\mathbf{C} - \mathbf{C}\cdot\mathbf{M} - \mathbf{B}\|_2^2 + \mu\psi(\mathbf{M}) + \lambda\phi(\mathbf{B}), \tag{3}$$

where $\psi(\mathbf{M})$ and $\mu$ are the regularization term and trade-off parameter for **M**. Due to the intrinsic ambiguity of foreground objects in concealed images and the diverse nature of their backgrounds, manually defining regularization terms for **M** and **B** can be challenging. To address this, we utilize deep neural networks to implicitly learn these constraints in a data-driven manner. Beyond the two intrinsic regularization terms above, we introduce an extra residual sparsity constraint $\mathcal{S}(\cdot)$ to further refine segmentation and minimize uncertainties. This leads to the final objective function:

$$\begin{aligned} L(\mathbf{M}, \mathbf{B}) = &\frac{1}{2}\|\mathbf{C} - \mathbf{C} \cdot \mathbf{M} - \mathbf{B}\|_2^2 + \mu\psi(\mathbf{M}) \\ &+ \lambda\phi(\mathbf{B}) + \alpha\mathcal{S}\left(\mathbf{w} \cdot \left(\mathbf{M} - \widetilde{\mathbf{M}}\right)\right), \end{aligned} \tag{4}$$

where $\alpha$ controls the weight of the sparsity constraint, $\mathcal{S}(\cdot)$ represents an $\ell_1$-norm, $\widetilde{\mathbf{M}}$ is the refined mask after uncertainty-removal mapping, and **w** is the attention map. For a pixel located at $(i, j)$, $\widetilde{\mathbf{M}}$ and **w** can be defined as

$$\widetilde{\mathbf{M}}_{(i,j)} = \begin{cases} 0.1 & \mathbf{M}_i \in [0.1, 0.4), \\ 0.9 & \mathbf{M}_i \in (0.6, 0.9], \\ \mathbf{M}_i & \text{Otherwise}. \end{cases} \tag{5}$$

$$\mathbf{w}_{(i,j)} = \begin{cases} 0 & \mathbf{M}_i \in [0.4, 0.6], \\ 1 & \text{Otherwise}. \end{cases}$$

This design encourages the generation of segmentation masks with high certainty. Following the practice of (He et al., 2024a), pixels with values in the ambiguous range $[0.4, 0.6]$ are excluded from further consideration, while extreme values for $\widetilde{\mathbf{M}}$ are set to 0.1 and 0.9 instead of 0 and 1 to allow greater flexibility for optimization.

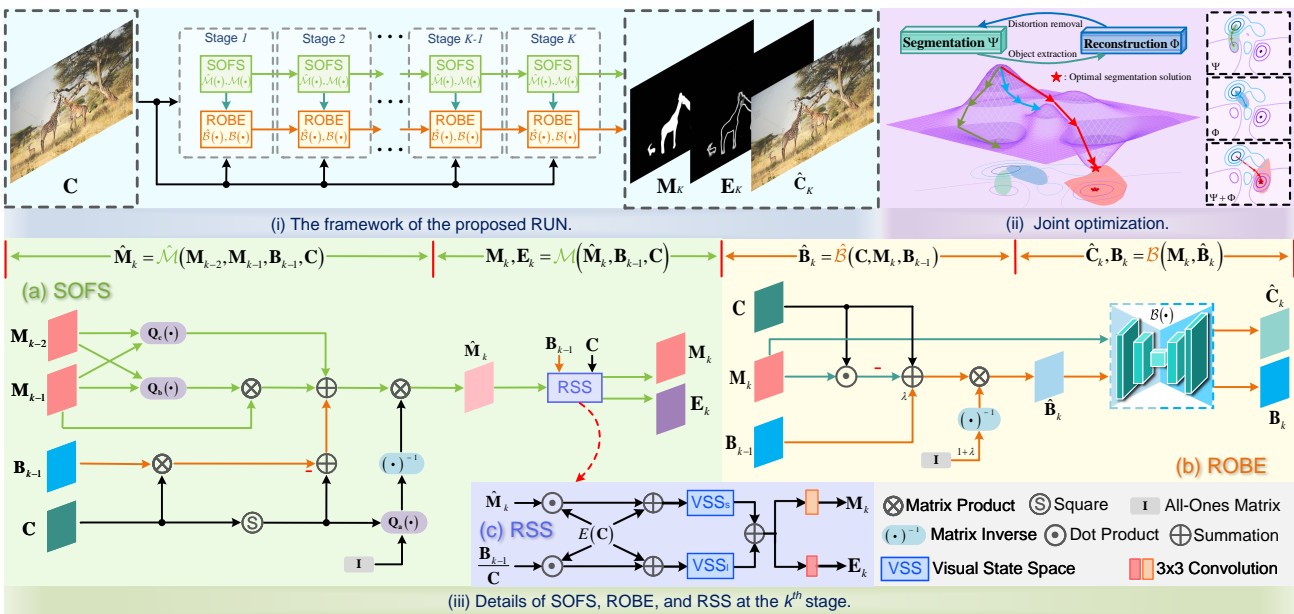

Figure 3. Framework of our RUN. The network connections in $\hat{\mathcal{M}}(\cdot)$ and $\hat{\mathcal{B}}(\cdot)$ are derived strictly based on mathematical principles, thus enhancing interpretability. For clarity, we replace certain redundant details with $\mathbf{Q_a}$, $\mathbf{Q_b}$, and $\mathbf{Q_c}$ and present $\hat{\mathcal{M}}(\cdot)$ according to Eq. (18). Panel (ii) illustrates that the joint optimization of image segmentation and reconstruction tasks facilitates the network's progression toward an optimal solution.

## 3.2. RUN

### 3.2.1. MODEL OPTIMIZATION

We utilize the proximal gradient algorithm (Fang et al., 2024) to optimize Eq. (4), ultimately deriving the optimal mask $\mathbf{M}^*$ and background $\mathbf{B}^*$:

$$\{\mathbf{M}^*, \mathbf{B}^*\} = \arg\min_{\mathbf{M}, \mathbf{B}} L(\mathbf{M}, \mathbf{B}). \quad (6)$$

The optimization process involves alternating updates of the two variables over iterations. Here we take the $k^{th}$ stage ($1 \leq k \leq K$) to present the alternative solution process.

**Optimizing $\mathbf{M}_k$.** First, the optimization function is partitioned to update the foreground mask $\mathbf{M}_k$:

$$\mathbf{M}_k = \arg\min_{\mathbf{M}} \frac{1}{2}\|\mathbf{C} - \mathbf{C} \cdot \mathbf{M} - \mathbf{B}_{k-1}\|_2^2 + \mu\psi(\mathbf{M})$$
$$+ \alpha\mathcal{S}\left(\mathbf{w}_k \cdot \left(\mathbf{M} - \widetilde{\mathbf{M}}_k\right)\right). \quad (7)$$

The solution comprises two terms: the gradient descent term and the proximal term. To address the proximal term, we introduce an auxiliary variable $\hat{\mathbf{M}}_k$, resulting in:

$$\mathbf{M}_k = \mathrm{prox}_\psi\left(\mathbf{B}_{k-1}, \hat{\mathbf{M}}_k\right), \quad (8)$$

where $\mathbf{B}_0$ is initialized as zero. Having gotten Eqs. (7) and (8), $\hat{\mathbf{M}}_k$ can be solved by optimizing:

$$\hat{\mathbf{M}}_k = \frac{1}{2}\|\mathbf{C} - \mathbf{C} \cdot \hat{\mathbf{M}} - \mathbf{B}_{k-1}\|_2^2 + \frac{\mu}{2}\|\hat{\mathbf{M}} - \mathbf{M}_{k-1}\|_2^2$$
$$+ \alpha\mathcal{S}\left(\mathbf{w}_k \cdot \left(\hat{\mathbf{M}} - \widetilde{\mathbf{M}}_k\right)\right), \quad (9)$$

where $\mathbf{M}_0$ is also initialized as zero. Note that $\mathbf{w}_k$ and $\widetilde{\mathbf{M}}_k$ are constructed based on $\mathbf{M}_{k-1}$. For the term $\mathcal{S}(\mathbf{w}_k \cdot (\hat{\mathbf{M}} - \widetilde{\mathbf{M}}_k))$, we employ a Taylor expansion rather than soft

thresholding for flexibility in problem-solving. Following the practice of (Goldstein, 1977), we approximate $\mathcal{S}(\mathbf{w}_k \cdot (\hat{\mathbf{M}} - \widetilde{\mathbf{M}}_k))$ at the $k - 1^{th}$ iteration (for simplicity, we let $\mathbf{R} = \mathbf{w}_k \cdot (\hat{\mathbf{M}} - \widetilde{\mathbf{M}}_k)$), expressed as follows:

$$\mathcal{S}(\mathbf{R}) \approx \dot{\mathcal{S}}(\mathbf{R}, \mathbf{R_{k-1}}), \quad (10)$$

where

$$\dot{\mathcal{S}}(\mathbf{R}, \mathbf{R_{k-1}}) \leftarrow \frac{L_\mathcal{S}}{2}\|\mathbf{R} - \mathbf{R}_{k-1} + \frac{1}{L_\mathcal{S}}\nabla\mathcal{S}(\mathbf{R}_{k-1})\|_2^2 + C_\mathcal{S}, \quad (11)$$

where $L_\mathcal{S}$ is the Lipschitz constant. $\nabla\mathcal{S}(\mathbf{R}_{k-1})$ is the Lipschitz continuous gradient function of $\mathcal{S}(\mathbf{R}_{k-1})$ with $C_\mathcal{S}$, a positive constant that can be omitted in optimization. Substituting into Eq. (9), we obtain the following equations:

$$\hat{\mathbf{M}}_k = \frac{1}{2}\|\mathbf{C} - \mathbf{C} \cdot \hat{\mathbf{M}} - \mathbf{B}_{k-1}\|_2^2 + \frac{\mu}{2}\|\hat{\mathbf{M}} - \mathbf{M}_{k-1}\|_2^2$$
$$+ \frac{\alpha L_\mathcal{S}}{2}\|\mathbf{R} - \mathbf{R}_{k-1} + \frac{1}{L_\mathcal{S}}\nabla\mathcal{S}(\mathbf{R}_{k-1})\|_2^2. \quad (12)$$

Unlike $\ell_1$-norm methods, Eq. (12) can be solved directly by equating its derivative to zero. The closed-form solution is

$$\hat{\mathbf{M}}_k = (\mathbf{Q_a})^{-1}\left(\mathbf{Q_b}\mathbf{M}_{k-1} + \mathbf{C}^2 - \mathbf{C}\mathbf{B_{k-1}} + \mathbf{Q_c}\right), \quad (13)$$

where $\mathbf{Q_a} = \mathbf{C}^2 + L_\mathcal{S}\mathbf{w}_k^2 + \mu\mathbf{I}$, $\mathbf{I}$ is an all-ones matrix, $\mathbf{Q_b} = \alpha L_\mathcal{S}\mathbf{w}_k\mathbf{w}_{k-1} + \mu\mathbf{I}$, $\mathbf{Q_c} = \alpha L_\mathcal{S}\mathbf{w}_k(\mathbf{w}_k \cdot \widetilde{\mathbf{M}}_k - \mathbf{Q_d}) - \alpha\mathbf{w}_k\nabla\mathcal{S}(\mathbf{w}_{k-1} \cdot \mathbf{M}_{k-1} - \mathbf{Q_d})$, and $\mathbf{Q_d} = \mathbf{w}_{k-1} \cdot \widetilde{\mathbf{M}}_{k-1}$.

**Optimizing $\mathbf{B}_k$.** The optimization function of $\mathbf{B}_k$ is

$$\mathbf{B}_k = \arg\min_{\mathbf{B}} \frac{1}{2}\|\mathbf{C} - \mathbf{C} \cdot \mathbf{M}_k - \mathbf{B}\|_2^2 + \lambda\phi(\mathbf{B}). \quad (14)$$

Same as the optimization rule for $\mathbf{M}_k$, the gradient descent term and the proximal term are correspondingly defined as:

$$\hat{\mathbf{B}}_k = \frac{1}{2}\|\mathbf{C} - \mathbf{C} \cdot \mathbf{M}_k - \hat{\mathbf{B}}\|_2^2 + \frac{\lambda}{2}\|\hat{\mathbf{B}} - \mathbf{B}_{k-1}\|_2^2, \quad (15)$$

$$\mathbf{B}_k = \text{prox}_\phi(\hat{\mathbf{B}}_k, \mathbf{M}_k). \tag{16}$$

The closed-form solution of $\hat{\mathbf{B}}_k$ can be acquired similarly:

$$\hat{\mathbf{B}}_k = ((1 + \lambda)\,\mathbf{I})^{-1} (\lambda \mathbf{B}_{k-1} + \mathbf{C} - \mathbf{C} \cdot \mathbf{M}_k). \tag{17}$$

### 3.2.2. DEEP UNFOLDING MECHANISM

We unfold the iterative optimization steps of the model-based solution into a multi-stage network, termed Reversible Unfolding Network (RUN), with each step corresponding to a stage. As shown in Fig. 3, each stage has two reversible modules: the Segmentation-Oriented Foreground Separation (SOFS) and Reconstruction-Oriented Background Extraction (ROBE) modules. The detailed variable update process for each stage is outlined in Algorithm S1.

**SOFS**. SOFS, derived from Eqs. (8) and (13), utilizes $\hat{\mathcal{M}}(\cdot)$ and $\mathcal{M}(\cdot)$ to compute the optimization solution $\hat{\mathbf{M}}$ and the refined mask $\mathbf{M}$ at each stage, respectively. Given $\mathbf{B}_{k-1}$, $\mathbf{M}_{k-1}$, and $\mathbf{M}_{k-2}$, we define $\hat{\mathbf{M}}_k$ as follows:

$$\begin{aligned}
\hat{\mathbf{M}}_k &= \hat{\mathcal{M}}(\mathbf{B}_{k-1}, \mathbf{M}_{k-1}, \mathbf{M}_{k-2}, \mathbf{C}), \\
&= (\mathbf{Q_a})^{-1}(\mathbf{Q_b}\mathbf{M}_{k-1} + \mathbf{C}^2 - \mathbf{C}\mathbf{B_{k-1}} + \mathbf{Q_c}).
\end{aligned} \tag{18}$$

Eq. (18) retains the same formulation as Eq. (13), but all originally fixed parameters, including $\nabla \mathcal{S}(\cdot)$, are relaxed to be learnable, improving the model's generalizability.

To refine the initial mask $\hat{\mathbf{M}}_k$, we introduce the Reversible State Space (RSS) module $RSS(\cdot)$, which has a robust capacity for non-local information extraction. The RSS module incorporates two Visual State Space (VSS) $VSS(\cdot)$ modules (Liu et al., 2024) with distinct perception fields. The VSS with a small perception field locally refines uncertain regions along the edges from the foreground perspective, while the VSS with a large perception field globally identifies missed segmented regions from the background perspective. This dual-perception mechanism ensures both accurate and comprehensive segmentation results. Following (He et al., 2023b), we also integrate an auxiliary edge output $\mathbf{E}_k$ to further enhance segmentation performance. Consequently, the computation of $\mathbf{M}_k$ and $\mathbf{E}_k$ is defined as:

$$\begin{aligned}
\mathbf{M}_k, \mathbf{E}_k &= \mathcal{M}(\mathbf{B}_{k-1}, \hat{\mathbf{M}}_k, \mathbf{C}) = RSS(\mathbf{B}_{k-1}, \hat{\mathbf{M}}_k, \mathbf{C}), \\
&= conv3(VSS_s(E(\mathbf{C}) \cdot \hat{\mathbf{M}}_k + E(\mathbf{C})) \\
&\quad + VSS_l(E(\mathbf{C}) \cdot (\mathbf{B}_{k-1}/\mathbf{C}) + E(\mathbf{C}))),
\end{aligned} \tag{19}$$

where $conv3$ is $3 \times 3$ convolution. $VSS_s(\cdot)$ and $VSS_l(\cdot)$ have small and large perception fields, incorporating convolutions with varying kernel sizes. For brevity, we omit the detailed description of VSS. Unlike low-level vision tasks, segmentation tasks, particularly inherently complex COS, strongly depend on semantic information. It is challenging to extract this fully using a shallow network. To address this, we adopt the common practice of leveraging deep features $E(\mathbf{C})$, extracted from an encoder (default: ResNet50 (He et al., 2016)). Rather than directly processing the concealed image, this approach enables the extraction of subtle dis-

criminative features, achieving accurate segmentation.

**ROBE**. In ROBE, the calculation of $\hat{\mathbf{B}}_k$ relies on $\hat{\mathcal{B}}(\cdot)$, similar to Eq. (15) but with the fixed parameters made learnable:

$$\begin{aligned}
\hat{\mathbf{B}}_k &= \hat{\mathcal{B}}(\mathbf{B}_{k-1}, \mathbf{M}_k, \mathbf{C}), \\
&= ((1 + \lambda)\,\mathbf{I})^{-1}(\lambda \mathbf{B}_{k-1} + \mathbf{C} - \mathbf{C} \cdot \mathbf{M}_k).
\end{aligned} \tag{20}$$

This is essentially a dynamic fusion of the previously estimated background and the reversed foreground derived in the current stage. To refine $\hat{\mathbf{B}}_k$, we propose a simple U-shaped network (Xu et al., 2023) with three layers, denoted as $\mathcal{B}(\cdot)$. However, as shown in Fig. 2, since separate modules estimate the foreground and background, their interpretations of the concealed content may differ. Hence, regions with conflicting interpretations are identified as distortion-prone areas in reconstruction. To address this, the network also generates a reconstructed result $\hat{\mathbf{C}}_k$, formulated as:

$$\mathbf{B}_k, \hat{\mathbf{C}}_k = \mathcal{B}\left(\hat{\mathbf{B}}_k, \mathbf{M}_k\right). \tag{21}$$

$\hat{\mathbf{C}}_k$ is designed to be consistent with the concealed image, thereby mitigating distortions. This alignment fosters consistent judgments between SOFS and ROBE for foreground-background separation, improving segmentation accuracy. As the stages progress, RUN incrementally facilitates reversible modeling of the foreground and background in both the mask and RGB domains. This iterative process directs the network's attention to regions of uncertainty, reducing false-positive and false-negative outcomes. Hence, RUN ensures robust and accurate segmentation performance.

**Loss function**. The loss function comprises a segmentation term and a reconstruction term. We adopt the training strategy from FEDER (He et al., 2023b) for the segmentation part. A mean square error loss governs the reconstruction component. The overall loss function is defined as

$$\begin{aligned}
L_t = \sum_{k=1}^{K} \frac{1}{2^{K-k}} & [L_{BCE}^w(\mathbf{M}_k, GT_s) + L_{IoU}^w(\mathbf{M}_k, GT_s) \\
& + L_{dice}(\mathbf{E}_k, GT_e) + \|\hat{\mathbf{C}}_k - \mathbf{C}\|_2^2],
\end{aligned} \tag{22}$$

where $K$ is the number of stages. $L_{BCE}^w$ is the weighted binary cross-entropy loss, $L_{IoU}^w$ is the weighted intersection-over-union loss, and $L_{dice}$ is the dice loss. $GT_s$ and $GT_e$ are the ground truth of the segmentation mask and edge.

## 4. Experiments

**Implementation details**. We implement our method using PyTorch on two RTX4090 GPUs. In line with (Fan et al., 2020a), we incorporate deep features from encoder-shaped networks into our framework. All images are resized to $352 \times 352$ for the training and testing phases. During training, we use the Adam optimizer with momentum parameters $(0.9, 0.999)$. The batch size is set to 36, and the initial learning rate is configured to 0.0001, which is reduced by 0.1 every 80 epochs. The stage number $K$ is set as 4. Additional

*Table 1.* Results on camouflaged object detection. SegMaR-1/-4 are SegMaR with one or four stages. The best results are marked in **bold**. For the ResNet50 backbone in the common setting, the best two results are in **red** and **blue** fonts.

| Methods | Backbones | CHAMELEON | | | | CAMO | | | | COD10K | | | | NC4K | | | |
|---|---|---|---|---|---|---|---|---|---|---|---|---|---|---|---|---|---|
| | | $M\downarrow$ | $F_\beta\uparrow$ | $E_\phi\uparrow$ | $S_\alpha\uparrow$ | $M\downarrow$ | $F_\beta\uparrow$ | $E_\phi\uparrow$ | $S_\alpha\uparrow$ | $M\downarrow$ | $F_\beta\uparrow$ | $E_\phi\uparrow$ | $S_\alpha\uparrow$ | $M\downarrow$ | $F_\beta\uparrow$ | $E_\phi\uparrow$ | $S_\alpha\uparrow$ |
| Common Setting: Single Input Scale and Single Stage | | | | | | | | | | | | | | | | | |
| SINet (Fan et al., 2020a) | ResNet50 | 0.034 | 0.823 | 0.936 | 0.872 | 0.092 | 0.712 | 0.804 | 0.745 | 0.043 | 0.667 | 0.864 | 0.776 | 0.058 | 0.768 | 0.871 | 0.808 |
| LSR (Lv et al., 2021) | ResNet50 | 0.030 | 0.835 | 0.935 | 0.890 | 0.080 | 0.756 | 0.838 | 0.787 | 0.037 | 0.699 | 0.880 | 0.804 | 0.048 | 0.802 | 0.890 | 0.834 |
| FEDER (He et al., 2023b) | ResNet50 | 0.028 | 0.850 | 0.944 | 0.892 | 0.070 | 0.775 | 0.870 | 0.802 | 0.032 | 0.715 | 0.892 | 0.810 | 0.046 | 0.808 | 0.900 | 0.842 |
| FGANet (Zhai et al., 2023) | ResNet50 | 0.030 | 0.838 | 0.945 | 0.891 | 0.070 | 0.769 | 0.865 | 0.800 | 0.032 | 0.708 | 0.894 | 0.803 | 0.047 | 0.800 | 0.891 | 0.837 |
| FocusDiff (Zhao et al., 2024) | ResNet50 | 0.028 | 0.843 | 0.938 | 0.890 | 0.069 | 0.772 | 0.883 | 0.812 | 0.031 | 0.730 | 0.897 | 0.820 | 0.044 | 0.810 | 0.902 | 0.850 |
| FSEL (Sun et al., 2024) | ResNet50 | 0.029 | 0.847 | 0.941 | 0.893 | 0.069 | 0.779 | 0.881 | 0.816 | 0.032 | 0.722 | 0.891 | 0.822 | 0.045 | 0.807 | 0.901 | 0.847 |
| RUN (Ours) | ResNet50 | 0.027 | 0.855 | 0.952 | 0.895 | 0.070 | 0.781 | 0.868 | 0.806 | 0.030 | 0.747 | 0.903 | 0.827 | 0.042 | 0.824 | 0.908 | 0.851 |
| BSA-Net (Zhu et al., 2022) | Res2Net50 | 0.027 | 0.851 | 0.946 | 0.895 | 0.079 | 0.768 | 0.851 | 0.796 | 0.034 | 0.723 | 0.891 | 0.818 | 0.048 | 0.805 | 0.897 | 0.841 |
| FEDER (He et al., 2023b) | Res2Net50 | 0.026 | 0.856 | 0.947 | 0.903 | 0.066 | 0.807 | 0.897 | 0.836 | 0.029 | 0.748 | 0.911 | 0.844 | 0.042 | 0.824 | 0.913 | 0.862 |
| RUN (Ours) | Res2Net50 | 0.024 | 0.879 | 0.956 | 0.907 | 0.066 | 0.815 | 0.905 | 0.843 | 0.028 | 0.764 | 0.914 | 0.849 | 0.041 | 0.830 | 0.917 | 0.859 |
| HitNet (Hu et al., 2023) | PVT V2 | 0.024 | 0.861 | 0.944 | 0.907 | 0.060 | 0.791 | 0.892 | 0.834 | 0.027 | 0.790 | 0.922 | 0.847 | 0.042 | 0.825 | 0.911 | 0.858 |
| CamoFocus (Khan et al., 2024) | PVT V2 | 0.023 | 0.869 | 0.953 | 0.906 | 0.044 | 0.861 | 0.924 | 0.870 | 0.022 | 0.818 | 0.931 | 0.868 | 0.031 | 0.862 | 0.932 | 0.886 |
| RUN (Ours) | PVT V2 | 0.021 | 0.877 | 0.958 | 0.916 | 0.045 | 0.861 | 0.934 | 0.877 | 0.021 | 0.810 | 0.941 | 0.878 | 0.030 | 0.868 | 0.940 | 0.892 |
| Other Setting: Multiple Input Scales (MIS) | | | | | | | | | | | | | | | | | |
| ZoomNet (Pang et al., 2022) | ResNet50 | 0.024 | 0.858 | 0.943 | 0.902 | 0.066 | 0.792 | 0.877 | 0.820 | 0.029 | 0.740 | 0.888 | 0.838 | 0.043 | 0.814 | 0.896 | 0.853 |
| FEDER (He et al., 2023b) | ResNet50 | 0.023 | 0.869 | 0.959 | 0.906 | 0.064 | 0.801 | 0.893 | 0.827 | 0.028 | 0.756 | 0.913 | 0.837 | 0.041 | 0.832 | 0.915 | 0.859 |
| RUN (Ours) | ResNet50 | 0.022 | 0.878 | 0.967 | 0.911 | 0.064 | 0.807 | 0.902 | 0.832 | 0.027 | 0.772 | 0.920 | 0.843 | 0.040 | 0.836 | 0.922 | 0.868 |
| Other Setting: Multiple Stages (MS) | | | | | | | | | | | | | | | | | |
| SegMaR-4 (Jia et al., 2022) | ResNet50 | 0.025 | 0.855 | 0.955 | 0.906 | 0.071 | 0.779 | 0.865 | 0.815 | 0.033 | 0.737 | 0.896 | 0.833 | 0.047 | 0.793 | 0.892 | 0.845 |
| FEDER-4 (He et al., 2023b) | ResNet50 | 0.025 | 0.874 | 0.964 | 0.907 | 0.067 | 0.809 | 0.886 | 0.822 | 0.028 | 0.752 | 0.917 | 0.851 | 0.042 | 0.827 | 0.917 | 0.863 |
| RUN-4 (Ours) | ResNet50 | 0.024 | 0.889 | 0.968 | 0.913 | 0.066 | 0.815 | 0.893 | 0.829 | 0.027 | 0.769 | 0.926 | 0.857 | 0.041 | 0.833 | 0.925 | 0.870 |

*Table 2.* Results on polyp image segmentation.

| Methods | CVC-ColonDB | | | ETIS | | |
|---|---|---|---|---|---|---|
| | mDice $\uparrow$ | mIoU $\uparrow$ | $S_\alpha\uparrow$ | mDice $\uparrow$ | mIoU $\uparrow$ | $S_\alpha\uparrow$ |
| PraNet (Fan et al., 2020b) | 0.709 | 0.640 | 0.819 | 0.628 | 0.567 | 0.794 |
| CASCADE (Rahman, 2023) | 0.809 | 0.731 | 0.867 | 0.781 | 0.706 | 0.853 |
| PolypPVT (Dong et al., 2023) | 0.808 | 0.727 | 0.865 | 0.787 | 0.706 | 0.871 |
| CoInNet (Jain et al., 2023) | 0.797 | 0.729 | 0.875 | 0.759 | 0.690 | 0.859 |
| LSSNet (Wang et al., 2024) | 0.820 | 0.741 | 0.867 | 0.779 | 0.701 | 0.867 |
| RUN (Ours) | 0.822 | 0.742 | 0.880 | 0.788 | 0.709 | 0.878 |

*Table 3.* Results on medical tubular object segmentation.

| Methods | DRIVE | | | CORN | | |
|---|---|---|---|---|---|---|
| | mDice $\uparrow$ | AUC $\uparrow$ | SEN $\uparrow$ | mDice $\uparrow$ | AUC $\uparrow$ | SEN $\uparrow$ |
| CS2-Net (Mou et al., 2021) | 0.795 | 0.983 | 0.822 | 0.607 | 0.960 | 0.817 |
| DSCNet (Qi et al., 2023) | 0.805 | 0.955 | 0.830 | 0.618 | 0.964 | 0.856 |
| SGAT (Lin et al., 2023) | 0.806 | 0.953 | 0.832 | 0.639 | 0.961 | 0.853 |
| TAU (Gupta et al., 2024) | 0.798 | 0.977 | 0.825 | 0.643 | 0.949 | 0.859 |
| FFM (Huang et al., 2025) | 0.791 | 0.972 | 0.830 | 0.647 | 0.952 | 0.835 |
| RUN (Ours) | 0.812 | 0.985 | 0.845 | 0.652 | 0.962 | 0.870 |

*Table 4.* Results on transparent object detection.

| Methods | GDD | | | GSD | | |
|---|---|---|---|---|---|---|
| | mIoU $\uparrow$ | $F_\beta^{max}\uparrow$ | $M\downarrow$ | mIoU $\uparrow$ | $F_\beta^{max}\uparrow$ | $M\downarrow$ |
| GDNet (Mei et al., 2020) | 0.876 | 0.937 | 0.063 | 0.790 | 0.869 | 0.069 |
| EBLNet (He et al., 2021) | 0.870 | 0.922 | 0.064 | 0.817 | 0.878 | 0.059 |
| RFENet (Fan et al., 2023b) | 0.886 | 0.938 | 0.057 | 0.865 | 0.931 | 0.048 |
| IEBAF (Han et al., 2024) | 0.887 | 0.944 | 0.056 | 0.861 | 0.926 | 0.049 |
| GhostingNet (Yan et al., 2024) | 0.893 | 0.943 | 0.054 | 0.838 | 0.904 | 0.055 |
| RUN (Ours) | 0.895 | 0.952 | 0.051 | 0.866 | 0.938 | 0.043 |

approaches across all three backbones: ResNet50 (He et al., 2016), Res2Net50 (Gao et al., 2019), and PVT V2 (Wang et al., 2022). This superior performance on four datasets, particularly on the largest dataset, *COD10K*, and the largest testing dataset, *NC4K*, underscores the robustness and generalization capabilities of our RUN framework. Furthermore, in the MIS and MS settings, our RUN adheres to the evaluation protocols of FEDER (He et al., 2023b) and delivers improved results over existing methods. As illustrated in Fig. 4, our method generates more complete and accurate segmentation maps. This is attributed to our jointly reversible modeling at both the mask and RGB levels.

**Medical concealed object segmentation**. We conducted experiments on two medical COS tasks, including polyp image segmentation (*CVC-ColonDB* and *ETIS* datasets) and medical tubular object segmentation (*DRIVE* and *CORN* datasets). Considering that recent SOTAs commonly use Transformer-based encoders, we adopt PVT V2 as our default encoder. As shown in Tables 2 and 3, our method achieves top performance across three tasks. Furthermore, the results in Fig. 4 confirm the effect of our approach in segmenting small polyps and fine vessels and nerves.

parameters inherited from traditional methods are optimized in a learnable manner with random initialization.

### 4.1. Comparative Evaluation

We conduct experiments on various COS tasks and compare our performance with SOTA methods using standard metrics. Details on datasets and metrics are in Appendix A.1. For fairness, all results are evaluated with consistent task-specific evaluation tools. Except for COD, other tasks have few publicly open-sourced methods, limiting quantitative analysis.

**Camouflaged object detection**. As shown in Table 1, our method achieves SOTA performance across all three settings. In the common setting, it outperforms competing

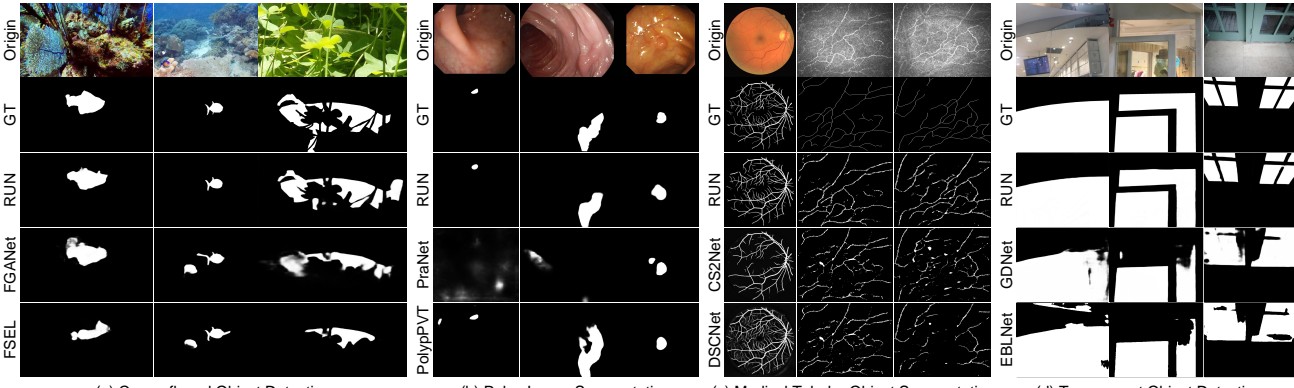

*Figure 4.* Visual comparison on COD, PIS, MTOS, and TOD tasks.

*Table 5.* Ablation study in the COD task on *COD10K*.

| Metrics | Effect of SOFS | | | | | | Effect of ROBE | | | Fixed → | RUN |
|---|---|---|---|---|---|---|---|---|---|---|---|
| | $\mathbf{C} \to E(\mathbf{C})$ | w/o RSS | w/o VSS | w/o prior $\hat{\mathbf{M}}_k$ | w/o prior $\mathbf{B}_{k-1}$ | w/o $\mathbf{E}_k$ | $\mathcal{B}_1(\cdot) \to \mathcal{B}(\cdot)$ | $\mathcal{B}_2(\cdot) \to \mathcal{B}(\cdot)$ | w/o $\hat{\mathbf{C}}_k$ | Learnable | (Ours) |
| $M \downarrow$ | 0.053 | 0.034 | 0.031 | 0.032 | 0.031 | 0.031 | 0.030 | 0.030 | 0.031 | 0.032 | 0.030 |
| $F_\beta \uparrow$ | 0.617 | 0.710 | 0.740 | 0.728 | 0.735 | 0.733 | 0.746 | 0.749 | 0.736 | 0.731 | 0.747 |
| $E_\phi \uparrow$ | 0.825 | 0.887 | 0.897 | 0.891 | 0.893 | 0.890 | 0.905 | 0.906 | 0.898 | 0.896 | 0.903 |
| $S_\alpha \uparrow$ | 0.746 | 0.805 | 0.825 | 0.820 | 0.826 | 0.823 | 0.826 | 0.828 | 0.824 | 0.822 | 0.827 |

*Table 6.* Effect of our COS model. CM, PM, OS, and DL are shorts for conventional model, proposed model, optimization solution, and deep learning.

| Metrics | PM+OS | CM1+DL | CM2+DL | CM3+DL | CM4+DL | CM5+DL | PM+DL (Ours) |
|---|---|---|---|---|---|---|---|
| $M \downarrow$ | 0.062 | 0.032 | 0.031 | 0.031 | 0.031 | 0.032 | 0.030 |
| $F_\beta \uparrow$ | 0.573 | 0.729 | 0.735 | 0.733 | 0.740 | 0.735 | 0.747 |
| $E_\phi \uparrow$ | 0.802 | 0.899 | 0.896 | 0.895 | 0.898 | 0.892 | 0.903 |
| $S_\alpha \uparrow$ | 0.733 | 0.823 | 0.824 | 0.821 | 0.823 | 0.823 | 0.827 |

*Table 7.* Analysis of stage number $K$. We have surpassed most compared methods when $K = 2$.

| Metrics | $K = 1$ | $K = 2$ | $K = 4$ (Ours) | $K = 6$ | $K = 8$ |
|---|---|---|---|---|---|
| $M \downarrow$ | 0.033 | 0.031 | 0.030 | 0.030 | 0.030 |
| $F_\beta \uparrow$ | 0.715 | 0.727 | 0.747 | 0.749 | 0.751 |
| $E_\phi \uparrow$ | 0.885 | 0.893 | 0.903 | 0.905 | 0.905 |
| $S_\alpha \uparrow$ | 0.803 | 0.812 | 0.827 | 0.826 | 0.830 |

**Transparent object detection**. Accurately segmenting transparent objects is crucial for autonomous driving. As demonstrated in Table 4 and Fig. 4, our RUN surpasses existing methods on two datasets, providing more precise segmentation of transparent objects compared to other approaches. These results highlight our potential to contribute to the advancement of autonomous driving.

### 4.2. Ablation Study

We conduct ablation studies on *COD10K* of the COD task.

**Effect of SOFS**. As presented in Table 5, replacing the deep features $E(\mathbf{C})$ with the concealed image results in performance decline, highlighting the critical role of incorporating deep features into the DUN-based framework. Additionally, we evaluate the impact of the state space-based structure by removing the RSS and VSS modules. The effectiveness of our reversible strategy is further validated by excluding the foreground prior $\hat{\mathbf{M}}_k$ and the background prior $\mathbf{B}_{k-1}$. Finally, we confirm the utility of integrating the auxiliary edge output, contributing to performance improvements.

**Effect of ROBE**. As shown in Table 5, when replacing $\mathcal{B}(\cdot)$ with other large-scale networks, *i.e.*, the CNN-based network $\mathcal{B}_1(\cdot)$ (Xu et al., 2023) and Transformer-based network $\mathcal{B}_2(\cdot)$ (He et al., 2025a), we observe no significant performance gains. This suggests that a simple network is

sufficient for background extraction and image reconstruction. Furthermore, when the reconstructed output $\hat{\mathbf{C}}_k$ is removed, our RUN also produces suboptimal results.

**Other configurations in RUN**. We validate the effect of various configurations in RUN. As shown in Table 5, allowing originally fixed parameters to be learnable enhances performance. Furthermore, we compare our model with CM1 to CM5, as detailed in Table 6. CM1 corresponds to Eq. (3). CM2 also employs Eq. (4) for optimization but applies the $\ell_1$-norm to the first term and employs soft thresholding (He et al., 2023a) to solve $\mathcal{S}(\cdot)$. CM3-CM5 represent ablated versions of the refined mask and weighted map in Eq. (4): CM3 removes the weighted map $\mathbf{w}$. CM4 modifies the refined range of pixel values from $[0.1, 0.4]\&(0.6, 0.9]$ to $[0.1, 0.3]\&(0.7, 0.9]$, with corresponding adjustments to the weighted map. CM5 extends the range to include all pixel values, assigning 0.5 to the foreground region. As shown in Table 6, our approach achieves superior performance across traditional solutions and learning-based unfolding strategies. Moreover, as verified in Table 7, we analyze the optimal stage number for our method. To balance performance and computational efficiency, we set $K = 4$. Under this configuration, the feature maps from the last four layers of the encoder are progressively sent to the four stages, with features from deeper layers transferred first.

Table 8. Results on small object images (1,084 images).

| Metrics | SegMaR | FEDER | FGANet | FocusDiff | FSEL | RUN (Ours) |
|---|---|---|---|---|---|---|
| $M \downarrow$ | 0.049 | 0.044 | 0.044 | 0.042 | 0.043 | **0.040** |
| $F_\beta \uparrow$ | 0.605 | 0.646 | 0.642 | 0.670 | 0.668 | **0.682** |
| $E_\phi \uparrow$ | 0.831 | 0.855 | 0.852 | 0.859 | 0.847 | **0.866** |
| $S_\alpha \uparrow$ | 0.765 | 0.777 | 0.776 | 0.781 | 0.776 | **0.789** |

Table 9. Results on multi-object images (186 images).

| Metrics | SegMaR | FEDER | FGANet | FocusDiff | FSEL | RUN (Ours) |
|---|---|---|---|---|---|---|
| $M \downarrow$ | 0.076 | 0.068 | 0.065 | 0.062 | 0.062 | **0.060** |
| $F_\beta \uparrow$ | 0.436 | 0.480 | 0.481 | 0.500 | 0.496 | **0.505** |
| $E_\phi \uparrow$ | 0.797 | 0.813 | 0.810 | 0.818 | 0.820 | **0.827** |
| $S_\alpha \uparrow$ | 0.695 | 0.709 | 0.709 | 0.716 | 0.717 | **0.730** |

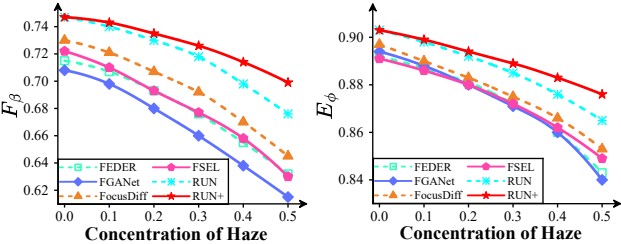

Figure 5. Performance in degraded COS scenarios.

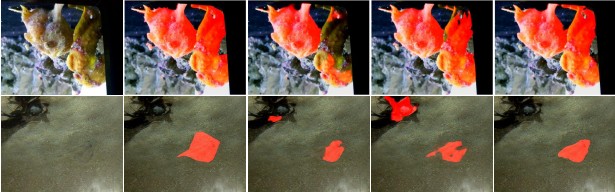

| (a) Origin | (b) GT | (c) FEDER | (d) FEDER-R | (e) FEDER+ |
|---|---|---|---|---|

Figure 6. Potential applications of RUN. The concealed object masks are highlighted in red and overlaid on the original images.

Table 10. Potential of RUN to serve as a refiner, where "FEDER-R" means refining FEDER's results with RUN.

| Metrics | FEDER | FEDER-R | FGANet | FGANet-R | FSEL | FSEL-R |
|---|---|---|---|---|---|---|
| $M \downarrow$ | 0.032 | 0.031 | 0.032 | 0.032 | 0.032 | 0.031 |
| $F_\beta \uparrow$ | 0.715 | 0.721 | 0.708 | 0.716 | 0.722 | 0.725 |
| $E_\phi \uparrow$ | 0.892 | 0.897 | 0.894 | 0.897 | 0.891 | 0.890 |
| $S_\alpha \uparrow$ | 0.810 | 0.812 | 0.803 | 0.805 | 0.822 | 0.825 |

Table 11. Generalization of RUN, where "FEDER+" means integrating our framework into FEDER.

| Metrics | FEDER | FEDER+ | FGANet | FGANet+ | FSEL | FSEL+ |
|---|---|---|---|---|---|---|
| $M \downarrow$ | 0.032 | 0.031 | 0.032 | 0.031 | 0.032 | 0.030 |
| $F_\beta \uparrow$ | 0.715 | 0.726 | 0.708 | 0.730 | 0.722 | 0.738 |
| $E_\phi \uparrow$ | 0.892 | 0.902 | 0.894 | 0.901 | 0.891 | 0.905 |
| $S_\alpha \uparrow$ | 0.810 | 0.816 | 0.803 | 0.808 | 0.822 | 0.830 |

## 4.3. Further Analysis, Applications, and Meanings

**Performance on small objects or multiple objects**. Small objects and multiple objects are challenging for lacking discriminative cues. To evaluate our performance on the two conditions, we filtered images from *COD10K* that satisfy these criteria, resulting in $1,084$ images having concealed objects smaller than a quarter of the entire image and 186 images with multiple concealed objects. As shown in Tables 8 and 9, while the performance of all methods declines, our approach consistently outperforms the competition.

**Performance on degraded COS scenarios**. To assess the impact of environmental degradation, we followed (He et al., 2023a) to simulate haze on concealed images in *COD10K* and then evaluated the ability of the compared methods to resist degradation. As illustrated in Fig. 5, performance degrades as the haze concentration increases. However, our RUN demonstrates superior resilience to haze degradation, attributed to its multi-modality reversible modeling strategy. To enhance robustness, we replaced our reconstruction network $\mathcal{B}(\cdot)$ with a more complex network from CoRUN (Fang et al., 2024), termed $\mathcal{B}_3(\cdot)$, which includes a pretrained dehazing model. This brought a novel unfolding network, RUN+, with $\mathcal{B}_3(\cdot)$ incorporating the pretrained model. Fig. 5 indicates integrating $\mathcal{B}_3(\cdot)$ enhances RUN's robustness in resisting haze degradation. This underscores the potential of RUN in addressing degraded scenarios.

**Potential applications of RUN**. First, we test the effect of our RUN as a refiner, specifically by initializing $\mathbf{M}_0$ with the results of existing methods. As shown in Table 10, our approach can enhance the performance of SOTA meth-

ods without requiring retraining. Furthermore, we incorporate the core structures of existing methods into our RUN framework, followed by retraining the entire network. This integration yields even greater improvements, demonstrating that the unfolding framework can function as a plug-and-play solution to enhance the performance of existing methods. For example, as shown in Fig. 6, we observe that while error predictions from FEDER influence FEDER-R, FEDER+ demonstrates better resilience to such errors.

**Meanings of our framework**. Beyond introducing the deep unfolding network to high-level vision for the first time and enabling reversible modeling across both mask and RGB domains, the proposed RUN framework offers the potential to establish a *unified vision strategy*. By combining image segmentation and image reconstruction, our RUN introduces a novel approach to unifying low-level and high-level vision. Unlike existing strategies, such as bi-level optimization (Xu et al., 2023; He et al., 2025d;c), our unfolding-based combination strategy is underpinned by explicit theoretical guarantees with the two models better coupled. Moreover, as shown in Fig. 5 RUN+, using more complex low-level vision algorithms results in a strong ability to resist complex degradation. This motivates further exploration of unfolding-based combination strategies to enhance high-level vision algorithms' resistance to environmental degradation and imaging interference. Simultaneously, it promotes low-level vision algorithms by integrating deep semantic information and high-level guidance. Together, these advancements ensure practical applicability in both real-world high-level and low-level vision tasks.

# 5. Conclusions

This paper proposes RUN to formulate the COS task as a foreground-background separation model. Its optimized solution is unfolded into a multistage network, where each stage comprises two reversible modules: SOFS and ROBE. SOFS applies the reversible strategy at the mask level and introduces RSS for non-local information extraction. ROBE employs a reconstruction network to address conflicting foreground and background regions in the RGB domain. Extensive experiments verify the superiority of RUN.

# Acknowledgements

This research was supported by the Foundation Fighting Blindness (BR-CL-0621-0812-DUKE); Research to Prevent Blindness (Unrestricted Grant to Duke University), Foundation Fighting Blindness (PPA-1224-0890-DUKE), and NSFC (No. 62476143).

# Impact Statement

This paper presents work whose goal is to advance the field of Machine Learning. There are many potential societal consequences of our work, none of which we feel must be specifically highlighted here.

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

---

**Algorithm S1** Proposed RUN Framework.

---

**Input**: concealed image $\mathbf{C}$, stage number $K$
**Output**: concealed object mask $\mathbf{M}_K$, concealed object edge $\mathbf{E}_K$, reconstructed concealed image $\hat{\mathbf{C}}_K$

1: Zero initialization for $\mathbf{M}_0$, $\mathbf{B}_0$
2: **for** each stage $k \in [1, K]$ **do**
3:     $\hat{\mathbf{M}}_k = \hat{\mathcal{M}}\left(\mathbf{B}_{k-1}, \mathbf{M}_{k-1}, \mathbf{M}_{k-2}, \mathbf{C}\right),$
4:     $\mathbf{M}_k, \mathbf{E}_k = \mathcal{M}(\mathbf{B}_{k-1}, \hat{\mathbf{M}}_k, \mathbf{C}),$
5:     $\hat{\mathbf{B}}_k = \hat{\mathcal{B}}\left(\mathbf{B}_{k-1}, \mathbf{M}_k, \mathbf{C}\right),$
6:     $\mathbf{B}_k, \hat{\mathbf{C}}_k = \mathcal{B}\left(\hat{\mathbf{B}}_k, \mathbf{M}_k\right).$
7: **end for**

---

## A. Experiment

### A.1. Datasets and metrics

**Camouflaged object detection**. In this task, we follow the standard practice of SINet (Fan et al., 2020a) and perform experiments on four datasets: *CHAMELEON* (Skurowski et al., 2018), *CAMO* (Le et al., 2019), *COD10K* (Fan et al., 2021a), and *NC4K* (Lv et al., 2021). The *CHAMELEON* dataset comprises 76 images, while the *CAMO* dataset contains 1,250 images divided into 8 classes. The *COD10K* dataset includes 5,066 images categorized into 10 super-classes, and *NC4K* serves as the largest test set, with 4,121 images. For training, we use 1,000 images from *CAMO* and 3,040 images from *COD10K*. The remaining images from these two datasets, along with all images from the other datasets, constitute the test set. To evaluate performance, we employ four widely-used metrics: mean absolute error ($M$), adaptive F-measure ($F_\beta$) (Margolin et al., 2014), mean E-measure ($E_\phi$) (Fan et al., 2021b), and structure measure ($S_\alpha$) (Fan et al., 2017). Superior performance is indicated by lower values of $M$ and higher values of $F_\beta$, $E_\phi$, and $S_\alpha$.

**Medical concealed object segmentation**. We evaluate the performance of our method on two specific tasks: polyp image segmentation and medical tubular object segmentation. For polyp image segmentation, we utilize two benchmarks: *CVC-ColonDB* (Tajbakhsh et al., 2015) and *ETIS* (Silva et al., 2014). The training protocol follows the setup of LSSNet (Wang et al., 2024). Quantitative evaluation is conducted using three commonly adopted metrics: mean Dice (mDice), mean Intersection over Union (mIoU), and structure measure ($S_\alpha$), where higher values indicate better performance. For medical tubular object segmentation, we evaluate our method on the *DRIVE*[1] and *CORN* (Ma et al., 2021) datasets, with training and inference conducted separately for each dataset. For the *DRIVE* dataset, training and inference adhere to the dataset's predefined splits. For the *CORN* dataset, the last $70\%$ of the data is used for training, while the first $30\%$ serves as the test set. Following DSCNet (Qi et al., 2023), we employ three evaluation metrics: mDice, area under the ROC curve (AUC), and sensitivity (SEN), with higher values reflecting better performance. To ensure a fair comparison with state-of-the-art medical concealed object segmentation methods, which predominantly utilize transformer-based encoders, we adopt PVT V2 as the backbone for our encoder.

**Transparent object detection**. For a fair comparison, we use PVT V2 as our default backbone and conduct experiments on two datasets: *GDD* (Mei et al., 2020) and *GSD* (Lin & He, 2021). The training set consists of 2,980 images from *GDD* and 3,202 images from *GSD*, while the remaining images are reserved for inference. Consistent with GDNet-B (Mei et al., 2023), we evaluate performance using several metrics, including mIoU, and maximum F-measure ($F_\beta^{max}$). Superior performance is indicated by lower values for $M$, or higher values for mIoU and $F_\beta^{max}$.

**Concealed defect detection**. In this task, we utilize PVT V2 as the default backbone. Consistent with established practices, we evaluate the generalization capacity of our RUN framework on the concealed defect detection task. Specifically, we use the model trained on the COD task to segment concealed objects in the *CDS2K* dataset (Fan et al., 2023a). Eight evaluation metrics are employed, where higher values indicate better performance for all metrics except MAE, for which lower values are preferred.

---

[1]http://www.isi.uu.nl/Research/Databases/DRIVE/

