# OpenReview forum: "RUN: Reversible Unfolding Network for Concealed Object Segmentation"
_ICML.cc/2025/Conference — ICML 2025 poster_

### Official Review · Reviewer_tcYD · 2025-03-07

**Overall Recommendation:** 3

**Summary:**

This paper tackles the challenging Concealed Object Segmentation, which aims to segment objects that are visually blended with their surroundings. It introduces the Reversible Unfolding Network (RUN), a iteratively method to refine segmentation results and minimize uncertainties. RUN includes two main modules: the Segmentation-Oriented Foreground Separation (SOFS) module, which captures non-local context and applies the reversible strategy at the mask level, and the Reconstruction-Oriented Background Extraction (ROBE) module, which addresses conflicting foreground and background regions. The method was evaluated on multiple datasets to valid its effectiveness.

**Claims And Evidence:**

The paper claims that previous works use reversible strategies but are generally limited to the mask domain. In contrast, this work introduces RUN, which extends reversible strategies to both the mask and RGB domains.

In P3L142, the paper states that integrating model-based and learning-based approaches remains underexplored due to the lack of intrinsic models. The paper should provide discussion to clarify the specific challenges involved and explain why addressing this gap is important.

**Essential References Not Discussed:**

[1] Bilateral Reference for High-Resolution Dichotomous Image Segmentation, CAAI AIR, 2024.

[2] ZoomNeXt: A Unified Collaborative Pyramid Network for Camouflaged Object Detection, TPAMI, 2024.

[3] SegRefiner: Towards Model-Agnostic Segmentation Refinement with Discrete Diffusion Process, NeurIPS, 2023.

**Experimental Designs Or Analyses:**

The methods is evaluated on multiple benchmarks using the standard metrics.

Although previous works have not fully explored the RGB domain for mask refinement, this method appears to underperform compared to BiRefNet [1] and ZoomNeXt [2]. For instance, the S-measure on CAMO (higher is better) is 0.806 for this work, whereas BiRefNet achieves 0.904. Additionally, the paper does not provide the comparison results with these methods.

The comparisons are insufficient. Please include a comparison with previous segmentation refinement methods, such as [3].

[1] Bilateral Reference for High-Resolution Dichotomous Image Segmentation, CAAI AIR, 2024.

[2] ZoomNeXt: A Unified Collaborative Pyramid Network for Camouflaged Object Detection, TPAMI, 2024.

[3] SegRefiner: Towards Model-Agnostic Segmentation Refinement with Discrete Diffusion Process, NeurIPS, 2023.

**Methods And Evaluation Criteria:**

The methods is evaluated on multiple benchmarks using the standard metrics.

According to Fig. 3, the architecture appears somewhat complex. The paper should provide a comparison of FPS and parameter count to better demonstrate the efficiency of this method.

**Other Comments Or Suggestions:**

N/A

**Other Strengths And Weaknesses:**

N/A

**Questions For Authors:**

Please refer to the comments in the above reviews.

**Relation To Broader Scientific Literature:**

This work proposes an iterative method to refine segmentation results, which may benefit related segmentation tasks.

**Theoretical Claims:**

N/A

---

> ### Author Rebuttal · Authors · 2025-03-31
>
> Thanks for the valuable comments.
>
> **W1. Challenges in introducing DUNs and importance of modeling high-level vision tasks**
>
> In P3L142, we state that deep unfolding networks (DUNs) are underexplored in high-level vision tasks for lacking intrinsic models.
>
> **Challenges:** DUNs rely on model-based optimization, which is effective in low-level vision tasks due to well-defined physical models (e.g., Retinex model for low-light enhancement, Atmospheric Scattering model for dehazing). **However, high-level vision tasks lack an explicit physical model, limiting DUN applications.**
>
> **Importance:**
> - Explicit constraints: Modeling high-level vision tasks enables task-specific constraints to guide optimization explicitly. For example, Eq. (4) introduces a residual sparsity constraint to refine segmentation and reduce uncertainties (P3L130, right part). This is more direct than implicit backpropagation. Table 5 shows a performance drop when removing this constraint (w/o prior $\hat{\mathbf{M}}_k$).
> - Introducing DUNs into high-level vision: With an explicit model, DUNs integrate optimization with deep networks, balancing interpretability and generalizability (P2L58, right part). RUN benefits in effectiveness (Table 1), efficiency (see **W2**), and degradation resistance (Fig. 5) in COS.
>
> **W2. Efficiency analysis**
>
> The designs of $\hat{\mathcal{M}}(\cdot)$ and $\hat{\mathcal{B}}(\cdot)$ follow mathematical principles and may appear complex, but we only make fixed hyperparameters learnable and enforce weight sharing for SOFS and ROBE across stages. In the table, our method is more efficient than SOTAs.
> |||Param. (M)/FLOPS (G)/FPS|
> |-|-|-|
> |ResNet50|FocusDiff|166.18/7618.49/0.23|
> ||RUN|30.41/43.36/22.75|
> |Res2Net50|FEDER|45.92/50.03/14.02|
> ||RUN|30.57/45.73/20.26|
> |PVT V2|CamoFocus|68.85/91.35/9.63|
> ||RUN|65.17/61.83/15.82|
>
> **W3. Comparison with more methods**
>
> BiRefNet [1] targets high-resolution segmentation (HRS); ZoomNeXt [2] focuses on multiple input scales (MIS), **differing from our setting**.
>
> Higher resolution and multiscale inputs enhance subtle discriminative cues but add computational overhead. We also report our results in these settings.
>
> For HRS, we retrain our method following BiRefNet [4], replacing our encoder with SwinL and training at 1024 x 1024 resolution. We evaluate our results on the three datasets reported in BiRefNet and find that RUN outperforms BiRefNet in most metrics. This is primarily due to our reversible modeling strategy.
> |HRS ($M$/$F_\beta$/$E_\phi$/$S_\alpha$)|CAMO|COD10K|NC4K|
> |-|-|-|-|
> |BiRefNet|0.030/0.895/0.954/0.904|0.014/0.881/0.960/0.913|0.023/0.925/0.953/0.914|
> |RUN|0.032/0.898/0.951/0.910|0.012/0.878/0.967/0.920|0.022/0.946/0.961/0.927|
>
> For MIS, we compare with ZoomNeXt (352 × 352). In the table, RUN outperforms ZoomNeXt in 8 out of 12 metrics. Unlike ZoomNeXt’s specialized extraction-fusion strategy, RUN simply concatenates multiscale features. Adding ZoomNeXt’s strategy to RUN yields RUN++, improving results.
> |MIS ($M$/$F_\beta$/$E_\phi$/$S_\alpha$)|CHAMELEON|CAMO|COD10K|NC4K|
> |-|-|-|-|-|
> |ZoomNext|0.020/0.872/0.963/0.912|0.069/0.782/0.883/0.822|0.026/0.765/0.918/0.855|0.038/0.827/0.919/0.869|
> |RUN|0.022/0.878/0.967/0.911|0.064/0.807/0.902/0.832|0.027/0.772/0.920/0.843|0.040/0.836/0.922/0.868|
> |RUN++|0.019/0.885/0.971/0.916|0.063/0.815/0.906/0.833|0.025/0.781/0.930/0.852|0.038/0.847/0.932/0.875|
>
> **W4. Performance in segmentation refinement**
>
> RUN, an end-to-end network for COS, can also serve as a segmentation refiner by initializing $\mathbf{M}_0$ as the coarse mask.
>
> We compare with several segmentation refiners: traditional methods (dense crf (DCRF) [3] and biliteral solver (BS) [4]) and learning-based methods (SegRefiner [5] and SAMRefiner [6]). To ensure fairness, we retrain the learning-based ones for COS, SegRefiner+ and SAMRefiner+, following their training rules.
>
> Traditional methods fail to refine FEDER’s segmentation in COD10K due to the challenges of COS (objects blending with surroundings). SegRefiner and SAMRefiner also perform suboptimally with their provided models, as coarse COS masks contain segmentation errors and uncertain regions, complicating refinement.
>
> Retraining improves SegRefiner+ and SAMRefiner+, with SAMRefiner+ getting results comparable to RUN. However, RUN (FPS: 22.75) outperforms SAMRefiner (FPS: 0.92) and SegRefiner (FPS: 0.89) in efficiency.
>
> Refinement results for FGANet’s masks are discussed in **W3** (reviewer 45Vr), with similar conclusions.
>
> ||FEDER|+DCRF|+BS|+SegRefiner|+SegRefiner+|+SAMRefiner|+SAMRefiner+|FEDER-R (Ours)|
> |-|-|-|-|-|-|-|-|-|
> |$M$|0.032|0.039|0.043|0.037|0.033|0.038|0.033|0.031|
> |$F_\beta$|0.715|0.683|0.666|0.691|0.718|0.686|0.723|0.721|
> |$E_\phi$|0.892|0.853|0.862|0.863|0.889|0.857|0.906|0.897|
> |$S_\alpha$|0.810|0.797|0.770|0.781|0.803|0.783|0.805|0.812|
>
> [1] BiRefNet, CAAI AIR24
>
> [2] ZoomNext, TPAMI24
>
> [3] Dense CRF, NIPS21
>
> [4] Bilateral Solver, ECCV16
>
> [5] SegRefiner, NIPS23
>
> [6] SAMRefiner, ICLR25

---

### Official Review · Reviewer_MZkc · 2025-03-09

**Overall Recommendation:** 5

**Summary:**

This paper proposes the first deep unfolding network, RUN, for the COS task, aiming to cope with one of the intrinsic challenges in the COS task, which is neglecting the importance of applying reversible strategies in the RGB domain. To achieve this goal, two reversible modules, SOFS and ROBE, are further proposed. By treating the foreground-background separation problem as the noise elimination problem, ROBE serves as a valuable assistant for SOFS in promoting accurate segmentation. Abundant experiments verify the potential of this method.

**Claims And Evidence:**

Yes

**Essential References Not Discussed:**

N/A

**Experimental Designs Or Analyses:**

The abundant experiments are both convincing and well-discussed.

**Methods And Evaluation Criteria:**

Yes

**Other Comments Or Suggestions:**

I think this paper is very interesting. Please consider addressing the weaknesses. I will consider modifying the score based on the response.

**Other Strengths And Weaknesses:**

Strengths:
The proposed RUN framework is not only a novel DUN-based application for the high-level vision task but also allows for the combination of high-level and low-level vision, ensuring robust performance in high-level vision tasks even in degraded scenarios.

Weaknesses:
1 The related work is very short. Please consider providing a more detailed description in the revised version.
2 Why use a very simple reconstruction network in ROBE, given its reconstruction capacity is highly limited in this condition?
3 Also, the reviewer wants to know if changing the simple network in ROBE to a more complex framework brings an evident performance gain.
4 The reviewer is curious about if the single SOFS module achieves comparable performance with existing methods.
5 More experiments are preferred to analyze the performance in complex degradation. The reviewer thinks this is a major challenge in COS. Please fully prove whether RUN can address this challenge and make comparison with existing strategies, such as bi-level optimization.

**Questions For Authors:**

See weaknesses.

**Relation To Broader Scientific Literature:**

This is the first application of deep unfolding network, a widely-used strategy in low-level vision tasks that is favored in achieving a balance in interpretability and generalizability, in high-level vision tasks. This brings positive effects to the whole high-level vision field.

**Theoretical Claims:**

The theoretical analysis in this paper is accurate.

---

> ### Author Rebuttal · Authors · 2025-03-31
>
> Thanks for the valuable comments.
>
> **W1. More related works**
>
> In related works, the COS component focuses on the development of the reversible modeling strategy, while our coverage of DUN is limited due to the underexploration of this technique in high-level vision tasks. To provide a more comprehensive overview, we will add a section that systematically reviews the advancements in the COS field.
>
> **W2&W3. Why use a simple network in ROBE? How about replacing it with a more complex one?**
>
> (1) The reconstruction task in ROBE is relatively easy, compared to most low-level vision tasks, as the input images are of high quality. Thus, we select a lightweight network to balance performance and computational efficiency. Experimental results validate the effectiveness of this design choice.
> |||Parameters (M)|FLOPS (G)|FPS|
> |-|-|-|-|-|
> |ResNet50|FocusDiff|166.18|7618.49|0.23|
> ||FSEL|29.15|51.03|2.19|
> ||RUN|30.41|43.36|22.75|
> |Res2Net50|FEDER|45.92|50.03|14.02|
> ||RUN|30.57|45.73|20.26|
> |PVT V2|CamoFocus|68.85|91.35|9.63|
> ||RUN|65.17|61.83|15.82|
>
> (2) As shown in Table 5, replacing the simple network with more complex networks brings limited performance gains. However, the results in Fig. 5 indicate that a more complex network can better resist degradation conditions.
>
> (3) Furthermore, as shown in the table, increasing the stage number can also improve the network’s degradation resistance capacity. These findings can help us further exploit the potential of this DUN-based framework.
> |$F_\beta$|0|0.1|0.2|0.3|0.4|0.5|
> |-|-|-|-|-|-|-|
> |RUN-4 stages|0.747|0.740|0.730|0.718|0.698|0.676|
> |RUN-6 stages|0.749|0.742|0.733|0.723|0.708|0.694|
> |RUN-8 stages|0.751|0.743|0.735|0.726|0.714|0.701|
>
> |$E_\phi$|0|0.1|0.2|0.3|0.4|0.5|
> |-|-|-|-|-|-|-|
> |RUN-4 stages|0.903|0.898|0.892|0.885|0.876|0.865|
> |RUN-6 stages|0.905|0.898|0.893|0.886|0.878|0.869|
> |RUN-8 stages|0.905|0.900|0.896|0.891|0.885|0.877|
>
> **W4. Performance of SOFS**
>
> As shown in the table, removing ROBE leads to a noticeable performance decline. The resulting model underperforms compared to the state-of-the-art method, FocusDiff, further highlighting the effect of our approach. This can also be verified by Table 11, where replacing SOFS with the core structures of existing methods results in clear performance improvements.
> ||w/o ROBE|RUN (Ours)|
> |-|-|-|
> |$M$|0.032|0.030|
> |$F_\beta$|0.713|0.747|
> |$E_\phi$|0.892|0.903|
> |$S_\alpha$|0.816|0.827|
>
> **W5. Compare with bi-level optimization on more degradation scenes**
>
> (1) As shown in Fig. 5, replacing the original simple network with a more complex dehazing network leads to RUN+, which improves robustness against haze degradation. To further evaluate our approach, we test its performance within a bi-level optimization (BLO) framework following the setup of HQG-Net [1], where ROBE and SOFS are treated as low-level and high-level vision blocks, as illustrated in Fig. S2. Evaluating performance on COD10K using $F_\beta$ and $E_\phi$, we find that RUN+ consistently outperforms the BLO framework across different haze concentrations.
> |$F_\beta$|0.1|0.2|0.3|0.4|0.5|
> |-|-|-|-|-|-|
> |BLO|0.741|0.732|0.720|0.705|0.697|
> |RUN+|0.743|0.735|0.726|0.718|0.708|
>
> |$E_\phi$|0.1|0.2|0.3|0.4|0.5|
> |-|-|-|-|-|-|
> |BLO|0.897|0.890|0.883|0.872|0.860|
> |RUN+|0.899|0.894|0.889|0.883|0.876|
>
> (2) Additionally, we further analyze the performance in two degradation scenarios that are extremely challenging for the COS task, i.e., the low-light and low-resolution scenarios.
>
> Low-light scenarios deepen the concealment of objects by lowering color contrast, while low-resolution scenes directly reduce the discriminative cues by reducing the number of valid pixels.
>
> As with the above content, we also simulate degradation on concealed images in COD10K.
>
> For the low-light scenarios, we applied image darkening [2] and degraded the dataset into three levels: slight low-light, medium low-light, and severe low-light. We employ Reti-Diff [3] as the reconstruction network in the ROBE module. Experiments verify that our RUN framework achieves better performance than the BLO version.
> |$F_\beta$ / $E_\phi$|slight|medium|severe|
> |-|-|-|-|
> |BLO|0.719/0.870|0.691/0.853|0.662/0.813|
> |RUN+|0.735/0.889|0.716/0.878|0.683/0.855|
>
> For the low-resolution scenes, we use bicubic downsampling and select x2, x4, x8 for reconstruction and segmentation. DiffIR [4] is selected as the reconstruction network. As shown in the table, RUN still outperforms BLO in the low-resolution challenges.
> |$F_\beta$/$E_\phi$|x2|x4|x8|
> |-|-|-|-|
> |BLO|0.706/0.865|0.662/0.847|0.580/0.786|
> |RUN+|0.729/0.881|0.707/0.866|0.625/0.820|
>
> To sum up, our RUN framework is not only a novel unfolding-based application for high-level vision tasks but also **effectively integrates high-level and low-level vision, ensuring robust performance even in degraded scenarios**.
>
> [1] HQGNet, TNNLS23
>
> [2] StableLLVE, CVPR21
>
> [3] Reti-Diff, ICLR25
>
> [4] DiffIR, ICCV23

---

> > ### Comment · Reviewer_MZkc · 2025-04-02
> >
> > The authors have resolved all my concerns, especially that the proposed RUN effectively bridges low-level and high-level tasks, ensuring robustness in degradation scenarios.
> >
> > It is indeed a strong piece of work. Given the inspired contribution and the comments from the other three reviewers, I am inclined to raise my score and recommend strong acceptance.

---

> > > ### Author Response · Authors · 2025-04-02
> > >
> > > We sincerely appreciate the reviewer’s recognition of the significance of our contribution. The RUN framework is not only a novel unfolding-based approach but also effectively integrates high- and low-level vision, ensuring robustness even in degraded scenarios.
> > >
> > > Your acknowledgment is highly valuable to us and reinforces our commitment to advancing research in this field!

---

### Official Review · Reviewer_45Vr · 2025-03-12

**Overall Recommendation:** 3

**Summary:**

The authors propose Reversible Unfolding Network (RUN) for Concealed Object Segmentation. RUN integrates optimization-based solutions with deep learning, enabling reversible modeling across both mask and RGB domains. It also introduces the Segmentation-Oriented Foreground Separation (SOFS) module and the Reconstruction-Oriented Background Extraction (ROBE) module. Extensive experiments validate its superiority.

**Claims And Evidence:**

N/A

**Essential References Not Discussed:**

N/A

**Experimental Designs Or Analyses:**

N/A

**Methods And Evaluation Criteria:**

N/A

**Other Comments Or Suggestions:**

N/A

**Other Strengths And Weaknesses:**

Strengths
- RUN develops the first deep unfolding network for COS, revealing the potential of deep unfolding network in COS.
- RUN introduces SOFS and ROBE modules to direct attention to uncertain regions through iterative optimization
- RUN achieves excellent performances across five different tasks against other task-specific methods.
- RUN can integrate with existing methods to further boost performance, showcasing its flexibility.

Weaknesses
- This paper does not have any discussion of computational efficiency. It can be found that the performance is worse than most SOTA methods when the stage number K is small by comparing the results in Table 7 and Table 1. However, a larger K requires longer inference time. The authors should compare the computational cost between RUN and other methods.
- Ablation studies are unclear. What is the baseline for this method? How does the model perform when the proposed SOFS and ROBE are removed?
- The proposed method can be regarded as a post-processing method. Therefore, it should be compared with common post-processing methods in segmentation, such as dense crf and bilateral solver.

**Questions For Authors:**

N/A

**Relation To Broader Scientific Literature:**

N/A

**Theoretical Claims:**

N/A

---

> ### Author Rebuttal · Authors · 2025-03-30
>
> Thanks for the valuable comments.
>
> **W1. Efficiency analysis**
>
> We compare the parameters, FLOPS, and FPS between our RUN against cutting-edge methods on three backbones. Our stage number is set to 4. As shown in the table, our method is more efficient across all three backbones with the input size of 352×352.
> |||Parameters (M)|FLOPS (G)|FPS|
> |-|-|-|-|-|
> |ResNet50|FocusDiff|166.18|7618.49|0.23|
> ||FSEL|29.15|51.03|2.19|
> ||RUN|30.41|43.36|22.75|
> |Res2Net50|FEDER|45.92|50.03|14.02|
> ||RUN|30.57|45.73|20.26|
> |PVT V2|CamoFocus|68.85|91.35|9.63|
> ||RUN|65.17|61.83|15.82|
>
> Besides, we analyze the computational cost of our RUN framework with varying stage numbers, along with their performance using ResNet50 as the backbone on COD10K. Since the SOFS and ROBE modules at different stages share the same weights, all versions of RUN maintain a consistent parameter count. As shown in the table, our method outperforms existing state-of-the-art methods once the stage number reaches 3. For an optimal balance between efficiency and performance, we set the stage number to 4.
> ||FLOPS (G)|FPS|$M$|$F_\beta$|$E_\phi$|$S_\alpha$|
> |-|-|-|-|-|-|-|
> |RUN-1 stage|29.95|28.61|0.033|0.715|0.885|0.803|
> |RUN-2 stages|34.12|25.27|0.031|0.727|0.893|0.812|
> |RUN-3 stages| 38.73|23.63|0.030|0.735|0.898|0.822|
> |RUN-4 stages| 43.36|22.75|0.030|0.747|0.903|0.827|
> |RUN-6 stages| 52.65|19.86|0.030|0.749|0.905|0.826|
> |RUN-8 stages| 62.02|16.53|0.030|0.751|0.905|0.830|
>
> **W2. Ablation study: baseline and more results**
>
> As shown in Table 5, we conduct ablation studies on the ResNet50-based RUN framework.
>
> To ensure the integrity of our reversible modeling strategy, we retain both SOFS and ROBE rather than performing breakdown ablations. Instead, in Table 5, we evaluate the contribution of each content within SOFS and ROBE by either replacing them with alternative designs or directly removing them, both of which lead to performance decreases.
>
> Here we report the results of removing SOFS and ROBE. To remove the segmentation module SOFS, we add an extra segmentation head, i.e., the convolution blocks, after ROBE, with the segmentation loss unchanged. For the case w/o ROBE, we simply remove ROBE and keep SOFS. Besides, we include a baseline where both SOFS and ROBE are removed, leaving only the segmentation head in each stage. As shown in the table, adding SOFS and ROBE brings a clear performance gain.
> ||Baseline|w/o SOFS|w/o ROBE|RUN (Ours)|
> |-|-|-|-|-|
> |$M$|0.053|0.038|0.032|0.030|
> |$F_\beta$|0.552|0.683|0.713|0.747|
> |$E_\phi$|0.718|0.855|0.892|0.903|
> |$S_\alpha$|0.682|0.797|0.816|0.827|
>
> **W3. Comparison with segmentation refinement methods**
>
> Our RUN is an end-to-end network designed for COS. For its structural characteristics, it can also be applied for refining coarse masks by initializing $\mathbf{M}_0$ as the coarse mask.
>
> We compare our method with segmentation refinement methods, including traditional methods (dense crf (DCRF) [1] and biliteral solver (BS) [2]) and learning-based methods (SegRefiner [3] and SAMRefiner [4]). We use segmentation results from FEDER and FGANet as coarse masks. For a fair comparison, we also provide retrained versions of the learning-based refiners for the COS task, SegRefiner+ and SAMRefiner+, following their original training rules.
>
> As shown in the table, traditional methods fail to improve the segmentation results of FEDER and FGANet. SegRefiner and SAMRefiner also achieve suboptimal performance with their provided models.
>
> This is because concealed object segmentation is an inherently difficult task, where objects are visually blended with their surroundings. Hence, unlike common segmentation tasks, this task provides far fewer discriminative cues for feature extraction. Besides, the coarse segmentation masks often suffer from significant quality issues, such as mis-segmentation, blurred edges, and high uncertainty in certain pixel regions, making mask refinement particularly challenging.
>
> After retraining, the learning-based refiners show noticeable performance gains, with SAMRefiner+ achieving results comparable to our RUN. However, our RUN framework achieves an FPS of 22.75, outperforming SAMRefiner with only 0.92 FPS.
> ||FEDER|+DCRF|+BS|+SegRefiner|+SegRefiner+|+SAMRefiner|+SAMRefiner+|FEDER-R (Ours)|
> |-|-|-|-|-|-|-|-|-|
> |$M$|0.032|0.039|0.043|0.037|0.033|0.038|0.033|0.031|
> |$F_\beta$|0.715|0.683|0.666|0.691|0.718|0.686|0.723|0.721|
> |$E_\phi$|0.892|0.853|0.862|0.863|0.889|0.857|0.906|0.897|
> |$S_\alpha$|0.810|0.797|0.770|0.781|0.803|0.783|0.805|0.812|
>
> ||FGANet|+DCRF|+BS|+SegRefiner|+SegRefiner+|+SAMRefiner|+SAMRefiner+|FGANet-R (Ours)|
> |-|-|-|-|-|-|-|-|-|
> |$M$|0.032|0.041|0.042|0.043|0.034|0.037|0.034|0.032|
> |$F_\beta$|0.708|0.665|0.649|0.662|0.698|0.682|0.725|0.716|
> |$E_\phi$|0.894|0.846|0.828|0.847|0.887|0.866|0.900|0.897|
> |$S_\alpha$|0.803|0.772|0.764|0.765|0.787|0.774|0.803|0.805|
>
> [1] Dense CRF, NeurIPS21
>
> [2] The fast bilateral solver, ECCV16
>
> [3] SegRefiner, NeurIPS23
>
> [4] SAMRefiner, ICLR25

---

> > ### Comment · Reviewer_45Vr · 2025-04-03
> >
> > The rebuttal responded to my questions accordingly, and I decide to raise the final rating to weak accept.

---

> > > ### Author Response · Authors · 2025-04-03
> > >
> > > We sincerely appreciate the reviewer’s recognition of the significance of our contribution. The RUN framework is not only a novel unfolding-based approach but also effectively integrates high- and low-level vision, ensuring robustness even in degraded scenarios.
> > >
> > > Your acknowledgment is highly valuable to us and reinforces our commitment to advancing research in this field!

---

### Official Review · Reviewer_BHcc · 2025-03-13

**Overall Recommendation:** 4

**Summary:**

The paper introduces a Reversible Unfolding Network (RUN), a novel deep unfolding network that integrates object segmentation and distortion restoration tasks. RUN combines a Segmentation-Oriented Foreground Separation (SOFS) module and a Reconstruction-Oriented Background Extraction (ROBE) module to achieve more accurate segmentation by reducing uncertainties. The framework allows reversible modeling of foreground and background in both the mask and RGB domains, focusing on uncertain regions to improve accuracy. Experimental results across various COS tasks demonstrate that RUN outperforms existing methods and provides flexibility for integration with other models.

**Claims And Evidence:**

This paper claims to be the first deep unfolding network for Concealed Object Segmentation (COS) problem. Based on the literature review of the past research works done for COS, this claim is clear and convincing.  Other than this claim, this paper is mainly positioned as an application paper, which optimize the objective function with new parameters and new architecture designs to tackle COS tasks. Therefore, the claim is specified for solving the particular COS problems among the given benchmarking datasets. Based on the experiments in the main paper and the supplementary materials, the proposed optimization techniques and networks are well supported.

**Essential References Not Discussed:**

N.A.

**Experimental Designs Or Analyses:**

The experiments have been conducted to compare the proposed method with a series of state of the art methods on the aforementioned 4 datasets. The evaluation metrics are selected properly.  For the ablation study, I can see only COD10K dataset is selected, is there any particular reason for not running it over all the datasets?

**Methods And Evaluation Criteria:**

The theoretical part of the proposed method is inspired by the observation that segmentation mask gradients should be sparse. It introduces an extra term in the objective function to enforce the certainty. The optimization of the objective function is achieved by implementing corresponding network and train the network based on the objective function loss. The methods are evaluated in 4 widely used benchmarking datasets : CHAMELEON, CAMO, COD10K, and NC4K. The baseline methods are selected from the recent state of the art methods in this task.  The proposed method consistently outperforms the baseline methods.

**Other Comments Or Suggestions:**

The authors are suggested to address the weakness mentioned above in the rebuttal period.

**Other Strengths And Weaknesses:**

Strengths
1. This paper is well written with thorough theoretical derivation, implementation details and experiment verification.
2. The supplementary material has provided extra complement information. The code is available and valid for generating the proper results claimed in the paper.

**Questions For Authors:**

1. What is the processing speed of the propose method compared to other state of the art methods? What is the trade-off and what is the best scenario for application.

**Relation To Broader Scientific Literature:**

The proposed method has contribution to the restricted area of image segmentation. The proposed optimization techniques can be applied to a broader scientific literature where the objective function has similar content in nature. The sparsity of gradients can be found in many fields including optical flow, denoising, etc.

**Theoretical Claims:**

I checked and verified the correctness of the equations from Eq. (4) to Eq. (17) for theoretical derivation of the proposed objective function.

---

> ### Author Rebuttal · Authors · 2025-03-30
>
> Thanks for the valuable comments.
>
> **W1. Why only conducting ablation studies on COD10K**
>
> Given that COD10K is a representative and high-quality dataset, we follow existing methods [1,2,3] to conduct ablation studies on COD10K. We have now included the results of ablation studies on three extra datasets. As shown in the table, the outcomes remain consistent with our initial conclusions. We will add this content to the revised manuscript.
> |||$\mathbf{C}\rightarrow E(\mathbf{C})$|w/o RSS|w/o VSS|w/o prior $\hat{\mathbf{M}}_k$|w/o prior $\mathbf{B}_{k-1}$|w/o$\mathbf{E}_k$|$\mathcal{B}_1(\cdot)\rightarrow\mathcal{B}(\cdot)$|$\mathcal{B}_2(\cdot)\rightarrow\mathcal{B}(\cdot)$|w/o $\hat{\mathbf{C}}_k$|Fixed$\rightarrow$Learnable|RUN (Ours)|
> |-|-|-|-|-|-|-|-|-|-|-|-|-|
> |CHAMELEON|$M$|0.053|0.030|0.028|0.029|0.027|0.028|0.027|0.026|0.029|0.029|0.027|
> ||$F_\beta$|0.764|0.837|0.842|0.832|0.845|0.849|0.859|0.861|0.845|0.837|0.855|
> ||$E_\phi$|0.885|0.938|0.943|0.936|0.945|0.948|0.953|0.956|0.944|0.940|0.952|
> ||$S_\alpha$|0.802|0.883|0.889|0.886|0.889|0.891|0.893|0.900|0.893|0.889|0.895|
> |CAMO|$M$|0.097|0.075|0.071|0.072|0.070|0.071|0.069|0.068|0.072|0.072|0.070|
> ||$F_\beta$|0.693|0.752|0.773|0.766|0.775|0.776|0.783|0.780|0.770|0.764|0.781|
> ||$E_\phi$|0.788|0.836|0.858|0.847|0.863|0.861|0.870|0.866|0.852|0.845|0.868|
> ||$S_\alpha$|0.723|0.785|0.799|0.790|0.796|0.801|0.805|0.808|0.802|0.798|0.806|
> |NC4K|$M$|0.072|0.049|0.043|0.046|0.043|0.043|0.043|0.042|0.045|0.046|0.042|
> ||$F_\beta$|0.735|0.801|0.819|0.800|0.820|0.817|0.826|0.825|0.816|0.807|0.824|
> ||$E_\phi$|0.823|0.889|0.903|0.898|0.900|0.902|0.904|0.907|0.902|0.897|0.908|
> ||$S_\alpha$|0.792|0.835|0.845|0.838|0.849|0.846|0.853|0.852|0.845|0.842|0.851|
>
> **W2. Efficiency, trade-off, and the best scenario of RUN**
>
> (1) Efficiency: We compare parameters, FLOPS, and FPS in three backbones and find that our RUN surpasses existing cutting-edge methods across all three backbones.
> |||Parameters (M)|FLOPS (G)|FPS|
> |-|-|-|-|-|
> |ResNet50|FocusDiff|166.18|7618.49|0.23|
> ||RUN|30.41|43.36|22.75|
> |Res2Net50|FEDER|45.92|50.03|14.02|
> ||RUN|30.57|45.73|20.26|
> |PVT V2|CamoFocus|68.85|91.35|9.63|
> ||RUN|65.17|61.83|15.82|
>
> (2) Structure-level trade-off: Although different stages in RUN share the same weights, the stage number still affects the overall computational cost. To analyze the trade-off, we investigate the impact of the stage number on our ResNet50-based framework. In the table, our method surpasses existing methods when the stage number reaches 3. For an optimal balance, we set the stage number to 4.
> ||FLOPS (G)|FPS|$M$|$F_\beta$|$E_\phi$|$S_\alpha$|
> |-|-|-|-|-|-|-|
> |RUN-2 stages|34.12|25.27|0.031|0.727|0.893|0.812|
> |RUN-3 stages| 38.73|23.63|0.030|0.735|0.898|0.822|
> |RUN-4 stages| 43.36|22.75|0.030|0.747|0.903|0.827|
> |RUN-6 stages| 52.65|19.86|0.030|0.749|0.905|0.826|
>
> (3) Framework-level trade-off: RUN achieves cutting-edge performance for the theoretical coupling of the segmentation module (SOFS) and the reconstruction module (ROBE). As shown in the table, ROBE brings an evident gain with limited costs. This can also be verified by Table 11, where replacing SOFS with the core structures of existing methods brings clear profits.  Ablations in Table 5 indicate that replacing the simple reconstruction network with more complex ones yields marginal gains. Hence, we opt for a simple network here.
> ||w/o ROBE|RUN (Ours)|
> |-|-|-|
> |$M$|0.032|0.030|
> |$F_\beta$|0.713|0.747|
> |$E_\phi$|0.892|0.903|
> |$S_\alpha$|0.816|0.827|
> |Parameters (M)|26.97|30.41|
> |FLOPS (G)|37.64|43.36|
>
> (4) The best application scenarios: RUN is not only a novel unfolding-based application for COS in segmenting clean concealed objects, but also has the potential to **ensure robust performance in degraded scenes**. As shown in Fig. 5, when replacing the reconstruction network with a more complex one, we get RUN+ and observe better hazy resistance capacity.
>
> To further verify this, we compare with bi-level optimization (BLO), a degradation-resistant framework. The differences between the two frameworks can be seen in Fig. S2, where our framework is superior in the theoretical coupling of the two models. We retrain our network (RUN+) in the BLO framework and report the results on COD10K with $F_\beta$ and $E_\phi$. We find that our RUN+ surpasses BLO across different hazy concentrations.
> |$F_\beta$/$E_\phi$|0.1|0.2|0.3|0.4|0.5|
> |-|-|-|-|-|-|
> |BLO|0.741/0.897|0.732/0.890|0.720/0.883|0.705/0.872|0.697/0.860|
> |RUN+|0.743/0.899|0.735/0.894|0.726/0.889|0.718/0.883|0.708/0.876|
>
> Besides, we analyze performance in two other degradation scenes, i.e., the low-light and low-resolution scenes, and observe that our RUN framework consistently surpasses BLO. For space limitations, we put the results in the response of **W5** of the reviewer MZkc. The reviewer can refer to the tables for more details.
>
> [1] PFNet, CVPR21
>
> [2] ZoomNet, CVPR22
>
> [3] FEDER, CVPR23

---

### Decision · Program_Chairs · 2025-05-01

**Decision:**

Accept (poster)

**Comment:**

After careful consideration of the reviewers' comments and the authors' rebuttal, I believe that the paper makes a meaningful contribution to the field of Concealed Object Segmentation. The novelty of the deep unfolding network approach, combined with the strong experimental results and robustness in degraded scenarios, justifies acceptance. While the framework's complexity and lack of baseline comparison are valid concerns, they do not outweigh the paper's strengths.